# PLLaVA : Parameter-Efficient LLaVA Extension from Image to Video Understanding

The video depicts a woman walking down a street at night. She is wearing a black leather jacket and sunglasses, and she is carrying a black purse. The street is wet, suggesting that it has recently rained. There are other people in the background, but they are not the focus of the video. The woman appears to be the main character, and she is walking with purpose. The overall atmosphere of the video is dark and moody.

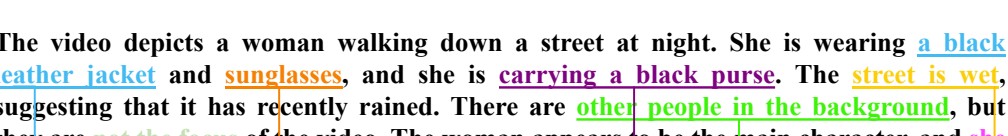

(a) PLLaVA generates dense descriptions of the video contents including motions, and attires.

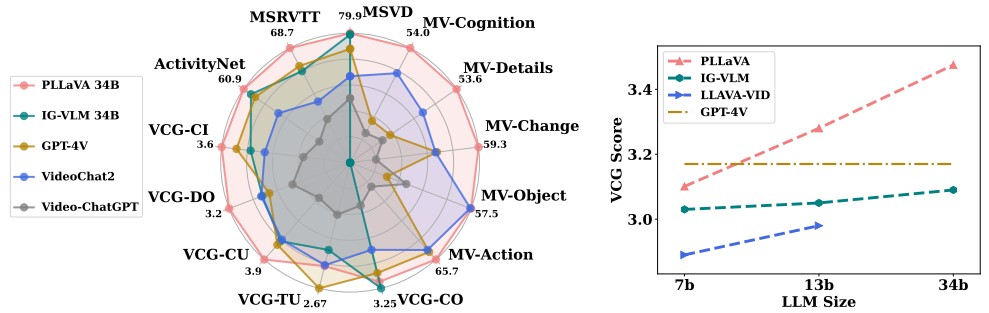

(b) Peformance on various video understanding tasks.

(c) Better model scaling performance.

Figure 1: Performance presentation of PLLaVA . (a) An example of captions generated with PLLaVA 34B. (b) Performance comparison of PLLaVA with recent strong baselines over different video benchmarks and (c) the scaling curve of PLLaVA and recent SOTA methods.

## Abstract

Vision-language pre-training has significantly elevated performance across a wide range of image-language applications. Yet, the pre-training process for video-related tasks demands exceptionally large computational and data resources, which hinders the progress of video-language models. This paper investigates a straight-forward, highly efficient, and resource-light approach to adapting an existing image-language pre-trained model for dense video understanding. Our preliminary experiments reveal that directly fine-tuning pre-trained image-language models with multiple frames as inputs on video datasets leads to performance saturation or even a drop. Our further investigation shows that it is largely attributed to the bias of learned high-norm visual features. Motivated by this finding, we propose a simple but effective pooling strategy to smooth the feature distribution along the temporal dimension and thus reduce the dominant impacts from the extreme features. The new model is termed Pooling LLaVA, or PLLaVA in short. PLLaVA achieves impressive performance on modern benchmark datasets for both video question-answer and captioning tasks. Notably, on the recent popular Video ChatGPT benchmark, PLLaVA achieves a score of 3.25 out of 5 on average of five evaluated dimensions. On the latest multi-choice benchmark MVBench, PLLaVA achieves 58.1% accuracy on average across 20 sub-tasks, 14.5% higher than GPT4V (IG-VLM). Our code is available at https://anonymous.4open.science/r/pllava_release-2B41.

## 1 Introduction

Multimodal Large Language Models (MLLMs) have demonstrated remarkable proficiency in image comprehension when trained on large-scale image-text pairs (23; 75; 34; 32; 17). Analogous to the image domain, the recent video understanding models also explore similar pipelines to fine-tune LLMs on large-scale video-text data (4; 24; 25). However, this method suffers a high cost of computing resources and video data annotations. A more pragmatic approach is to *adapt* the pre-trained image-domain MLLMs to video data (40; 36; 20). In this paper, without crafting too much data source and format, we investigate the model structures and training strategies to improve the understanding abilities of video LLMs.

An intuitive method of adapting image MLLMs into video domain is to directly encode multiple video frames to visual tokenks into MLLMs, as Large Language Models(LLMs) (51; 50) are native for processing sequential features and shown to be capable of understanding temporal information (29; 37). However, we empirically found two technical challenges when extending image MLLMs to the video domain in this way based on the existing public video-text data: **i)** *Training the image MLLM with video domain data does not always increase performance but introduces performance vulnerability to the change of inquiry prompts.* **ii)** *Increasing the size of the language model component does not improve the video understanding performance.* Those two observations are counter-intuitive since scaling up model sizes and exposing models to more downstream data are typically considered beneficial for model performance.

We then conducted a series of studies to investigate the root cause of these two observations. **For the data scaling challenge, we found it is mainly due to the limited and imbalanced information encoded by the visual encoder.** When experimenting on LLaVA (34) with 4-frame inputs, we found that, as shown in Figure 2(a), some learned visual tokens exhibit *dominantly larger norms* compared to others, suggesting two issues of these visual features: a) The information representation is uneven, which overemphasize on some types of information, e.g. the global video information, while suppressing other tokens containing the local detail details; b) these visual tokens in a whole contain less information according to the theory of information entropy (46). These tokens lead to shorter text descriptions with lower quality. As demonstrated in Figure 2(b), the 4-frame models tend to generate shorter texts with training on more samples. Even worse, if the prompt template changes, the learned MLLMs would completely collapse, leading to rather short descriptions or even no response. This worse performance with data scaling is due to the increasingly uneven visual features caused by the softmax operation in the self-attention with more training.

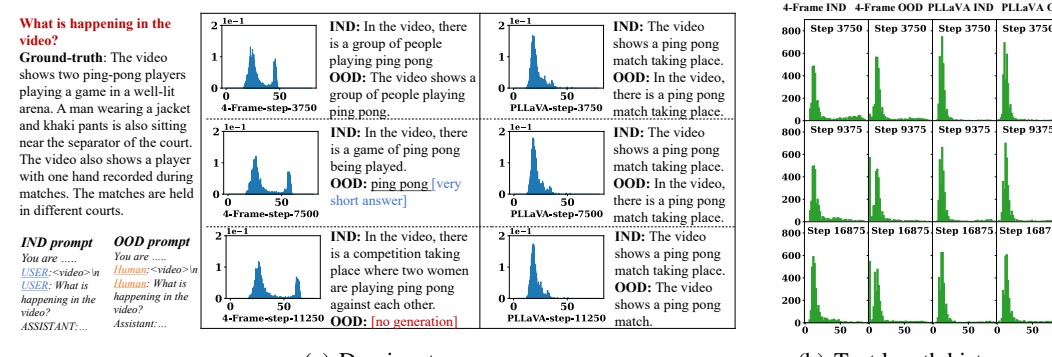

(a) Dominant norms.  (b) Text length histograms.

Figure 2: (a) An example comparing the token embedding norm distributions and generated texts of the *4-Frame* method and PLLaVA . For the *4-Frame* setting, from top to bottom, dominant tokens (with high norms) are more prevalent and show larger norm values(wider distance between two peaks), as more data samples are trained. This is accompanied by a decline in generation quality, particularly with out-of-distribution prompts. In the right column, PLLaVA presents consistent norm distributions and generated texts across various amounts of training data and prompts. (b) Histograms of generated text lengths for the *4-Frame* method and PLLaVA . The x-axis is text lengths, and the y-axis is the frequency of each text length. The *4-Frame* method generates shorter texts with more training steps and under out-of-distribution prompts, whereas PLLaVA maintains consistent text lengths in both situations.

This decline in performance with increasing data is attributed to the growing unevenness in visual features, a result of the softmax operation in self-attention. We show the preliminary proof in Sec. 4.2. Adding more video frames could be a potential solution to provide more information in the visual tokens, but this would lead to significantly larger memory consumption.

Considering the trade-off between information richness and the computation cost, an intuitive way is to downsample the video features. However, directly averaging the spatial and temporal dimensions as has been done in VideoChatGPT(40) loses too much spatial information and also does not achieve optimal performance during the scaling of the training dataset. Thus, our target is to find the minimum spatial resolution of each frame that does not degrade the scaling curve. To achieve this, we adopt a pooling (21) operation to explore the optimal settings such that it does not degrade the benefits of increasing the temporal receptive field. The impact of the pooling operation is shown in Figure 5.

**For the model size scaling issue, we believe one primary reason is the poorer quality of the applied video datasets compared to that of the image domain**. Specifically, many video datasets contain only simple video captions and are in question-answering format, often featuring brief answer descriptions. As the model learns the temporal descriptions from the video dataset, the describing ability of other metrics such as the objects and the spatial relations degrades. Additionally, our findings reveal a correlation: the stronger an LLM is, the quicker its output quality deteriorates under these circumstances.

Instead of building high-quality video datasets, we choose to explore architectural and optimization algorithms to better preserve the learned vision understanding and text generation ability in image datasets during the learning of the temporal information on video datasets. To achieve this, we utilize the tricks of weight fusion. We set two groups of weights: one from the image pre-raining and one with video dataset fine-tuning. After training, we searched to find the optimal combination of the image-based model weights and the video-based model weights in the hope that the combined model could gather benefits from both datasets. The process is termed post-training optimization in this paper and its impacts are shown in Figure 3(c). In a summary,

- We performed a thorough initial investigation for directly applying image large multi-modality models to video tasks and found several failure modes. We then introduce an elegantly simple yet highly potent pooling strategy that systematically achieves the optimal balance between training efficiency and understanding ability.

- We introduce a post-training model merging method that could effectively reduce the forgetting phenomenon of the large language models during multi-modality fine-tuning.

With this, we are able to get a large video multi-modality model with 34B LLMs without the extra creation of high-quality datasets.

- We conduct extensive experiments to verify the superiority of the proposed model and achieve some state-of-the-arts across various video understanding benchmarks, especially for video captioning tasks with dense captions. With PLLaVA , we do the re-captioning of 1K samples from the Inter4K (48) with highly dense and accurate bilingual captions.

## 2 RELATED WORKS

The success of image LLMs has encouraged studies in video LLMs. Various techniques are investigated to advance the video understanding abilities of LLMs.

**Parameter-Efficient Video Understanding.** One track of studies is dedicated to connecting video inputs and text outputs through a small number of parameters or adapting directly from image MLLMs to video understanding. Commonly, they incorporate a projection network (40; 30; 27; 16), inter-modality attention (24; 25) or a modality perceiver (70; 47; 18) as learnable interfaces. These interfaces are instrumental in melding the spatial-temporal dynamics of videos with large language models' (LLMs) processing capabilities (50; 45; 8), by transforming video content into a sequence of tokens that LLMs can adeptly analyze. Similar to BLIP2 (23), VideoLLaMA (70), Vista-LLaMA (39), VideoChat (24) and its advanced version VideoChat2 (25) employed cross-attention mechanisms to encode the input video tokens, ensuring a fixed amount of input context length. These methods align user queries with the dialogue context to enhance the model's interpretative capabilities. VideoChat2 is exceptional with a multi-stage bootstrapping technique that honed in on modality alignment and instruction tuning. Video-LLaVA (30) and CAT (64) resorted to ImageBind (13) to extract text-compatible video features, benefiting from fusion multi-modality data. However, a more efficient way to adapt image MLLMs for videos. Video-ChatGPT (40), on the other hand, directly extracted compressed spatial and temporal features with image MLLMs and reused the LLM part for text generation. IG-VLM (20) adapted the image MLLMs into the video domain by transforming videos into grid view images and SF-LLaVA (61) adopts two granurity when dealing with video frames. However, these methods could cause severe information loss due to improper feature compression and reduced frame resolution. TC-LLaVA (10) introduce a new position encoding method to emphasize the video frame locations. For additional related work on recent video multi-modal large language models (MLLMs), please refer to Appendix A.

## 3 METHOD & ANALYSIS

Adapting image MLLMs to the video domain can be challenging and susceptible to the designs of model structures, given the limited performance of existing methods.

### 3.1 FAILURE CASES ANALYSIS FOR APPLYING IMAGE MLLMs

We first explored a direct way to adapt image MLLMs into the video domain: concatenate visual tokens from several video frames as the input to image MLLMs. This approach leverages the LLMs' capability to interpret temporal information from the video frames. We termed this method as *n-frame*. Formally, given a sequence of video frames $\mathbf{X} \in \mathbb{R}^{T \times C \times W \times H}$, we obtain the features for each frame via the vision encoder pre-trained in CLIP-ViT (42) models. The encoded frame features are represented as $X_v \in \mathbb{R}^{T \times w \times h \times d}$. The MLLM then generates responses as follows:

$$r = \text{MLLM}(X_v, X_t), \tag{1}$$

where $X_t$ is the text input and $r$ is the output. However, two issues prevented us from achieving optimal performance in our attempts to train the MLLM with this method.

**Vulnerability to prompts.** The first observation is that the *n-frame* model is highly sensitive to prompt patterns when handling generation tasks. Figure 2(a) illustrates this phenomenon. We divide the prompts into two categories: in-distribution (IND) and Out-of-Distribution (OOD), the former is the prompt used during training while the latter is modified in format but has the same meaning. In the left part of the figure, when using IND, the model can generate decent video despite its tendency of shorter generation length with more data samples trained. However, when applying OOD prompts,

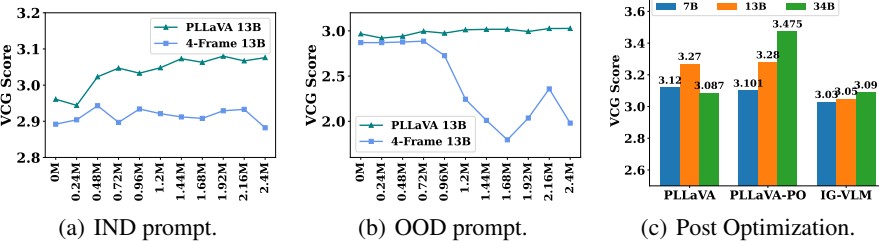

(a) IND prompt.  (b) OOD prompt.  (c) Post Optimization.

Figure 3: Validation curves for the *4-Frame* method and PLLaVA are displayed. In (a), the curves are shown for the in-distribution (IND) prompt, while (b) shows the curves for the out-of-distribution (OOD) prompts. The *4-Frame* method saturates quickly and even declines with prolonged training, whereas PLLaVA continues to improve. In (c), it is demonstrated that Video MLLMs fail to improve with increased model size, but Post Optimization effectively resolves the scaling degradation.

the quality of the generated responses drastically declines. The generation has content in normal length under the model trained for 3750 steps. However, for the longer trained models, the generations are shorter under 7500 steps, and no response under 11250 steps.

**Dominant tokens.** Given the previously mentioned vulnerability of *n-frame* models, we proceeded to analyze the variance between models at both their initial and fully-trained stages. By visualizing the norm of vision tokens across models at various training stages, we observed a trend towards the emergence of dominant tokens (characterized by high norms) as the number of training samples increased, as illustrated by the histograms in Figure 2(a). Additionally, the distribution of token norms became more pronounced with additional training data, indicating an increase in the norm of high-norm tokens. Consequently, we speculate that there is a plausible correlation between these dominant tokens and the degradation in generation quality with more data training. Comparisons of the distributions between the *n-frame* model and the proposed PLLaVA further support this conjecture, as detailed in Sec. 4.4.

**Difficulty to improve with more data.** Data scaling has been a widely accepted means to improve the LLMs' capability. However, The above phenomena indicate that employing image MMLMs in the video domain and seeking to benefit from the scaling of video data samples raises a challenging issue. We present *n-frame*'s performance(the blue curve) under dif-

| Method | Video-ChatGPT | | |
|---|---|---|---|
| | reported | reproduce | scaled |
| Dataset | 100K | 100K | 100K+249K |
| VCG Score | 2.38 | 2.41 | 1.94 |

Table 1: Video-ChatGPT (40) fails in data scaling.

ferent training samples in Figure 3. This figure illustrates that *n-frame* keeps stagnant under IND prompt, and degrades a lot under OOD prompts after the training sample exceeds 0.48M. Similar patterns are observed in the experimental findings of Video-ChatGPT (40), as detailed in Table 1. Video-ChatGPT (40) introduces a unique pooling strategy that involves averaging visual features across the temporal dimension as well as the spatial dimension, resulting in a visual feature $X_{vcg} \in \mathbb{R}^{(T+w \times h) \times d}$ after concatenating both dimensions. This feature is then fed into LLMs to generate a corresponding response. The first two columns of Table 1 demonstrate our replication of Video-ChatGPT using their 100K video-text dataset, while the third column illustrates a significant deterioration in model performance upon introducing additional training video data samples from VideoChat2 (25). **Consequently, identifying effective model strategies to exploit the ever-increasing amount of data to reach the data scaling law remains a critical issue.**

## 3.2 MODEL SCALING DEGRADATION

Our investigation of current video models reveals that it is also not straightforward to benefit models from scaled parameter sizes. We draw the performance of a recent work IG-VLM (20) and our attempts in Figure 3(c). IG-VLM achieves almost no difference when applying 7B, 13B, and 34B models of LLaVA-Next (33). In our initial attempts with pooled video features (the first column of Figure 3(c)), the experimental results on LLaVA-Next 34B are even worse than the 13B model. For IG-VLM, the input video frames are combined into a grid view image, confined by the resolution, leading to the unsatisfactory model size scaling ability. As for our initial attempts, we found a tendency of shorter generations with larger MLLMs, we thus owe the degradation to the quality of video-text data, which undermines the generation ability of LLMs in MLLM models.

## 3.3 PLLaVA

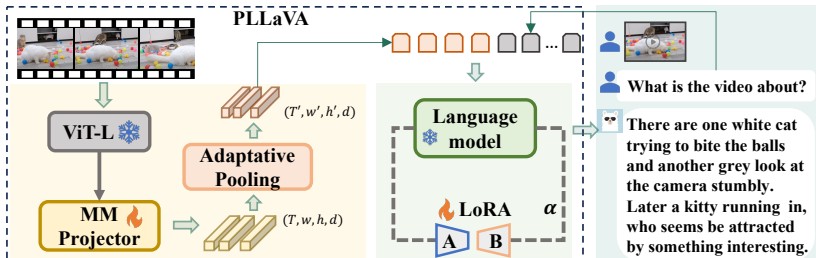

Figure 4: The framework of PLLaVA begins with processing a video from the user through ViT-L and MM projector, yielding visual features with shape $(T, w, h, d)$. These features undergo average pooling, which effectively reduces both temporal and spatial dimensions. The pooled features are then flattened and concatenated with question embeddings, serving as input to the image Large Language Model to generate a response to the user. The weights of the image LLMs are fused with LoRA weight learned under video samples.

**Motivation.** Our study of *n-frame* and VideoChatGPT (40) highlights the challenges of adapting image-based MLLM to the video domain. Notably, these two methods employ fundamentally different strategies for processing video inputs. The former utilizes a limited number of video frames, whereas the latter compresses over 100 frames using an averaging technique. Given the importance of temporal information and the high computational cost of MLLM inputs, pooling emerges as an intuitive and efficient solution to balance these needs. The challenges may arise from insufficient frame information and suboptimal processing of frame features. Motivated by these insights, we investigate the video feature pooling strategies employed in MLLM.

**Definition.** We formalize the pooling process for video features as follows: As shown in Figure 4, after feeding video frames $\mathbf{X} \in \mathbb{R}^{T \times C \times W \times H}$ into the CLIP-ViT model and the multimodal projector, we obtain an encoded vision feature $X_v \in \mathbb{R}^{T \times w \times h \times d}$ for a video input, where $T$ is the frame numbers, $C, W, H$ are the channel number, width and height of a frame, and $w, h, d$ are the dimensions of features. This feature is then passed through a parameter-free Adaptive Average Structure Pooling module[1] and reduced to a smaller dimensions $T' \times w' \times h'$, formulated as:

$$X_{vp} = \text{AdaptStructPooling}(X_v | T' \times w' \times h'). \tag{2}$$

These features are fed into LLMs with text input embeddings to generate responses. We also include a LoRA (15) module to adapt the LLM to video-related generation tasks. In conclusion, the trainable weights include Multimodal Projector and LLM LoRA. Within this framework, we investigated the impact of pooling through grid search analysis. Our findings suggest that pooling on the spatial dimension yields favorable outcomes, whereas temporal dimension pooling is associated with decreased performance. For a thorough exploration of our search process and the rationale behind this conclusion, please refer to Sec. 4.2.

**Pooling Effects.** Our experiments show that pooling to introduce more video frames can relieve dominant tokens. We provide preliminary theoretical proof to explain the underlying reasons. Generally, our conclusion is that dominant tokens, characterized by high token embedding norms, arise from sharply distributed inputs and are further amplified by the softmax operation. The pooling over more video frames promotes more balanced numerical input distributions, thereby mitigating the presence of dominant tokens.

The softmax function, converts a vector of values into a probability distribution. The softmax function softmax : $\mathbb{R}^n \to [0, 1]^n$ for a vector $z \in \mathbb{R}^n$ is defined as:

$$\text{softmax}(z)_i = \frac{e^{z_i}}{\sum_{j=1}^n e^{z_j}}, \text{ for } i = 1, 2, \ldots, n \tag{3}$$

We also define different input distributions: 1) Balanced Distribution $B$: A vector where elements $b_i$ are fairly close to each other, not necessarily uniform but without extreme deviations. 2) Sharp Distribution $S$: A vector with a significant outlier, $s_k$, much larger than the other components $s_j$ for $j \neq k$. We can look at the derivatives of the softmax function components with

---

[1] https://pytorch.org/docs/stable/generated/torch.nn.AdaptiveAvgPool3d.html

respect to its inputs: $\frac{\partial}{\partial z_i}\text{softmax}(z)_i = \text{softmax}(z)_i(1 - \text{softmax}(z)_i)$ and $\frac{\partial}{\partial z_j}\text{softmax}(z)_i = -\text{softmax}(z)_i\text{softmax}(z)_j$ $(i \neq j)$. Note that these derivatives typically show that that softmax outputs are more sensitive to changes at indices where $\text{softmax}(z)_i$ is larger.

When applying the softmax function to: 1) Sharp Distribution $S$: If $s_k \gg s_j$ for $j \neq k$, then $\text{softmax}(S)_k$ can be approximated to 1, and $\text{softmax}(S)_j$ for $j \neq k$ to 0. Any small perturbation in $s_k$ or $s_{j \neq k}$ will significantly alter the non-dominating probabilities $\text{softmax}(S)_j$. 2) Balanced Distribution $B$: Variations in $b_i$ cause smoother and smaller proportional changes in $\text{softmax}(B)_i$ since no single component overly dominates the exponential sum in the denominator. The probability distribution remains more uniform, and changes in one component slightly tweak the probabilities without extreme jumps.

Consequently, for a sharp distribution, due to the extreme values making one of the exponents dominantly larger, a tiny change in input can cause substantial shifts in some of the output probabilities. Larger outputs (due to significant input features) cause substantial gradients. The optimizer adjusts these weights more prominently compared to others. Eventually, these enlarged weight cause part the of learned feature larger, thus leading to dominant token embeddings. Conversely, in a balanced distribution, changes in inputs lead to proportional and smoother adjustments in probabilities, ensuring more stable outputs.

### 3.4 POST OPTIMIZATION

Regarding the difficulty model size scaling, which may stem from diminished language proficiency due to training on low-quality video-text data samples as stated in Sec. 3.2. To retain the language ability, we propose a post-training optimization(stated as Post Optimization from here) approach for video MLLMs. It blends the trained LLM weights on video data with the original LLM of the base image MLLM. Specifically, for a pretrained MLLM with LLM parameters $W_0$ and the input vision feature $X_{vp}$, the output hidden states after Post Optimization is defined as:

$$h = W_0 X_{vp} + \frac{\alpha}{r}\Delta W X_{vp}, \tag{4}$$

where $\Delta W$ are low-rank learnable parameters for $W_0$, and $\frac{\alpha}{r}$ is used to scale the learned low-rank weight. In Post Optimization, we tune the mix ratio between the original LLMs and the trained LLMs (incorporating LoRA weights) by varying the value of $\alpha$ during inference. Our experiments indicate that lower $\alpha$ yields significantly better generative performance. The larger $\alpha$ used accelerates the training phase while smaller $\alpha$ ensures better language ability during inference.

## 4 EXPERIMENTS

### 4.1 EXPERIMENT SETTING

**Data and Evaluation.** We leverage instructional video-to-text datasets to adapt image MLLMs to video inputs. The training data are sourced from VideoChat2 (25), which embraces data for various video understanding tasks, including 27k conversation videos from VideoChat (24) and Video-ChatGPT (40), 80k data of classification tasks from Kinetics (19) and SthSthV2 (11), 450k captioned data from Webvid (2), YouCook2 (74), TextVR (58) and VideoChat, 117 reasoning data from NextQA (59) and CLEVRER (65) and 109K annotated questioning answering data samples from Webvid, TGIF (28) and Ego4D (12). In total, we use 783k instructional tuning data.

We evaluate video LLMs with the following video-to-text benchmarks. First, the open-ended Video Question Answer (VideoQA) includes MSVD-QA (60), MSRVTT-QA (60), ActivityQA (67), and TGIF QA (28). Responses in these question-answering benchmarks are typically single word. GPT-3.5 (41) is used to evaluate the accuracy (Accuracy, with answers true/false) and quality (Score, ranging from 0 to 5) of the models' responses. Additionally, we adopt the Video-based Generative Performance benchmark (referred to as VCG Score) to measure generation performance, introduced by VideoChatGPT (40). This benchmark involves longer answers, encompassing five aspects of video understanding: CI (Correctness of Information), DO (Detail Orientation), CU (Context Understanding), TU (Temporal Understanding), and CO (Consistency). The benchmark also relies on GPT-3.5 model for assessments. Furthermore, we include the multi-choice Question Answering benchmark, MVBench (25), comprising 20 tasks that demand nuanced temporal comprehension of videos. This benchmark does not necessitate evaluation from the GPT-3.5 model.

**Models and Implementation Details.** We leverage pre-trained image MLLM weights from the Huggingface library and incorporate average pooling to reduce feature dimensions before feeding the input visual features into the LLM generation component. For the pooling layer, we uniformly sample 16 frames as input and define the target pooling shape as $16 \times 12 \times 12 \times d$, where $d$ represents the input dimension of the LLMs. During training, we use a batch size of 128, a learning rate of 2e-5, a cosine scheduler, and a warmup ratio of 0.03. All reported results are based on models trained for 6250 steps. For evaluation, we utilize the GPT-3.5-turbo-0125 model for evaluation across all benchmarks. Our experiments were conducted on a maximum of 16 A100 GPUs, requiring approximately 72 hours to complete training the 34B model for a single epoch.

## 4.2 IMPACT OF POOLING OPERATION DESIGN

Considering the unsatisfying performance of the complete pooling on temporal and spatial dimensions adopted by Video-ChatGPT, and the limitation information used in the *n-frame* method, we explore the influence of pooling strategies here.

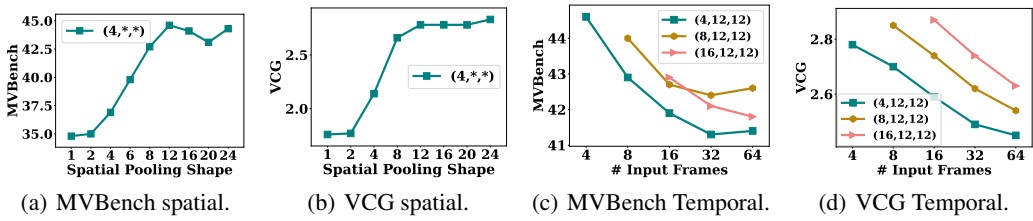

(a) MVBench spatial. (b) VCG spatial. (c) MVBench Temporal. (d) VCG Temporal.

Figure 5: Pooling shape influence.

**Pooling Layer Design** Pooling can be done both temporally and spatially for video features. We want to answer two questions: 1) which dimension is more suitable to be pooled? and 2) what is the largest compression ratio along that dimension? We plot a model curve based on the LLaVA-1.5 7B model with different temporal and spatial dimensions.

For the spatial dimension, we picked an input video feature with shape (4,24,24,$d$), where 4 is the frame numbers (temporal dimension), 24×24 is the original spatial dimension of frame features, and $d$ is the embedding dimension of each visual token. The target spatial shapes are chosen at evenly spaced intervals between 1 and 24, resulting in a set of spatial shapes $S =\{n \times n$ — $n \in [1, 2, 4, 6, 8, 12, 16, 20, 24]\}$. The MVBench and VCG Score performance of these spatial pooling shapes are shown in Figure 5(a) and 5(b). It is observed that downsampling the spatial dimension by 50% does not degrade the model performance. Further reducing the spatial dimension would lead to a significant performance drop. Considering the tradeoff between computational overhead and performance, 12×12 is chosen.

For the temporal dimension, several target pooling shapes were chosen with spatial dimensions fixed as 12, including (4,12,12), (8,12,12), and (16,12,12). We study the temporal pooling effects by altering the number of input video frames. For example, pooling from (64,24,24) to (4,12,12) indicates every 16 frames are fused, then the downsampling rate should be 6.25%. All of the resulting model curves are shown in Figure 5(c) and 5(d). Different from spatial pooling, the model performance is sensitive to temporal pooling. As illustrated in these two figures, all lines achieve better performance with lower downsampling rates. In other words, *pooling along temporal dimension always downgrades the model performance.*

**Pooling Impact.** We found that pooling over more video frames not only improves the model efficiency but also makes the model more robust to user inquiries. During our experiments, we evaluated models under different training iterations with two sets of prompts. For example, we vary the role tag from 'USER' to 'Human' during evaluation and the results are as shown in Figure 2(a). The figure shows that the visual feature norms learned with the pooling operation present consistent distributions under different training iterations compared to the *4-frame* method that shows dominant tokens. This is also reflected in the model responses where the pooling method gives consistent good text responses while the 4-frames method gives shorter and shorter text responses as the training goes longer, or even no response when out-of-distribution prompts are used. This conclusion can be further validated by Figure 2(b). With pooling introduced, no matter what prompt is used or how much training sampled is learned, the text generation lengths with the pooling method are consistent. We owe the stability in the generation to the smoothing ability of pooling, which eliminates the

influence of dominant high norm tokens. However, we haven't done a more rigorous analysis from the perspective of mathematical proofs, we leave it for future work.

## 4.3 QUANTITATIVE RESULTS

| Method | Vision Encoder | LLM Size | MSVD-QA | | MSRVTT-QA | | ActivityNet-QA | | TGIF-QA | | Video-ChatGPT | | | | | |
|---|---|---|---|---|---|---|---|---|---|---|---|---|---|---|---|---|
| | | | Acc. | Sco. | Acc. | Sco. | Acc. | Sco. | Acc. | Sco. | CI | DO | CU | TU | CO | Avg. |
| FrozenBiLM(62) | ViT-L | 1.3B | 33.8 | - | 16.7 | - | 25.9 | - | 41.9 | - | | | | | | |
| Video-LLaMA(70) | CLIP-G | 7B | 51.6 | 2.5 | 29.6 | 1.8 | 12.4 | 1.1 | - | - | 1.96 | 2.18 | 2.16 | 1.82 | 1.79 | 1.98 |
| LLaMA-Adapter(71) | ViT-B | 7B | 54.9 | 3.1 | 43.8 | 2.7 | 34.2 | 2.7 | - | - | 2.03 | 2.32 | 2.30 | 1.98 | 2.15 | 2.16 |
| Video-ChatGPT(40) | ViT-L | 7B | 64.9 | 3.3 | 49.3 | 2.8 | 35.2 | 2.7 | 51.4 | 3.0 | 2.50 | 2.57 | 2.69 | 2.16 | 2.20 | 2.42 |
| Video-LLaVA(30) | ViT-L | 7B | 70.7 | 3.9 | 59.2 | 3.5 | 45.3 | 3.3 | 70.0 | 4.0 | | | | | | |
| Chat-UniVi(18) | ViT-L | 7B | 65.0 | 3.6 | 54.6 | 3.1 | 45.8 | 3.2 | 60.3 | 3.4 | 2.89 | 2.91 | 3.46 | 2.89 | 2.81 | 2.99 |
| MovieChat(47) | CLIP-G | 7B | 75.2 | 3.8 | 52.7 | 2.6 | 45.7 | 3.4 | - | - | 2.76 | 2.93 | 3.01 | 2.24 | 2.42 | 2.67 |
| VideoChat(24) | CLIP-G | 7B | 56.3 | 2.8 | 45.0 | 2.5 | 26.5 | 2.2 | 34.4 | 2.3 | 2.23 | 2.50 | 2.53 | 1.94 | 2.24 | 2.29 |
| VideoChat2(25) | UMT-L | 7B | 70.0 | 3.9 | 54.1 | 3.3 | 49.1 | 3.3 | - | - | 3.02 | 2.88 | 3.51 | 2.66 | 2.81 | 2.98 |
| Vista-LLaMA(39) | CLIP-G | 7B | 65.3 | 3.6 | 60.5 | 3.3 | 48.3 | 3.3 | - | - | 2.44 | 2.64 | 3.18 | 2.26 | 2.31 | 2.57 |
| LLaMA-VID(27) | CLIP-G | 13B | 70.0 | 3.7 | 58.9 | 3.3 | 47.5 | 3.3 | - | - | 2.96 | 3.00 | 3.53 | 2.46 | 2.51 | 2.89 |
| LITA(17) | CLIP-L | 7B | - | - | - | - | - | - | - | - | 2.94 | 2.98 | 3.43 | 2.68 | 3.19 | 3.04 |
| ST-LLM(37) | BLIP2 | 7B | 74.6 | 3.9 | 63.2 | 3.4 | 50.9 | 3.3 | - | - | 3.23 | 3.05 | 3.74 | 2.93 | 2.81 | 3.15 |
| IG-VLM CogAgent(14) | CLIP-E | 7B | 76.7 | 4.1 | 62.7 | 3.6 | 57.3 | 3.6 | 76.7 | 4.0 | 3.26 | 2.76 | 3.57 | 2.34 | 3.28 | 3.04 |
| IG-VLM LLaVA 7B (33) | ViT-L | 7B | 78.8 | 4.1 | 63.7 | 3.5 | 54.3 | 3.4 | 73.0 | 4.0 | 3.11 | 2.78 | 3.51 | 2.44 | 3.29 | 3.03 |
| IG-VLM LLaVA 13B (33) | ViT-L | 13B | 77.4 | 4.1 | 62.6 | 3.4 | 57.1 | 3.5 | 78.0 | 4.0 | 3.17 | 2.79 | 3.52 | 2.51 | 3.25 | 3.05 |
| IG-VLM LLaVA 34B (33) | ViT-L | 34B | 79.6 | 4.1 | 62.4 | 3.5 | 58.4 | 3.5 | 79.1 | 4.2 | 3.21 | 2.87 | 3.54 | 2.51 | **3.34** | 3.09 |
| VILA 1.5 40B(31) | InternViT | 40B | **80.1** | - | 63 | - | 58 | - | 58.2 | - | - | - | - | - | - | - |
| TC-LLaVA 7B(10) | ViT-L | 7B | 78.8 | 4.1 | 63.2 | 3.6 | 56.8 | 3.5 | 78.2 | 4.2 | 3.25 | 2.96 | 3.75 | 2.91 | 3.09 | 3.19 |
| IG-VLM GPT-4V(1) | Unk | GPT-4 | 76.3 | 4.0 | 63.8 | 3.5 | 57.0 | 3.5 | 65.3 | 3.7 | 3.40 | 2.80 | 3.61 | **2.89** | 3.13 | 3.17 |
| PLLaVA 7B | ViT-L | 7B | 76.6 | 4.1 | 62.0 | 3.5 | 56.3 | 3.5 | 77.5 | 4.1 | 3.21 | 2.86 | 3.62 | 2.33 | 2.93 | 3.12 |
| PLLaVA 13B | ViT-L | 13B | 75.7 | 4.1 | 63.2 | 3.6 | 56.3 | 3.6 | 77.8 | 4.2 | 3.27 | 2.99 | 3.66 | 2.47 | 3.09 | 3.27 |
| PLLaVA 34B | ViT-L | 34B | 79.9† | **4.2†** | **68.7** | **3.8** | **60.9†** | **3.7†** | **80.6** | **4.3** | **3.60†** | **3.20†** | **3.90†** | 2.67† | 3.25 | **3.25†** |
| Improve over GPT-4V (20) | - | - | 3.6 | 0.2 | 4.9 | 0.3 | 3.9 | 0.2 | 15.3 | 0.6 | 0.2 | 0.4 | 0.3 | -0.32 | 0.12 | 0.31 |

Table 2: Results of video question-answering. † indicates our PLLaVA 34B significantly outperforms IG-VLM LLaVA 34B under the t-test and Wilcoxon test, with p-values close to 0.0. Values without † are because of our lower performance or the missing results from IG-VLM.

| Method | Vision Encoder | LLM Size | AS | AP | AA | FA | UA | OE | OI | OS | MD | AL | ST | AC | MC | MA | SC | FP | CO | EN | ER | CI | Avg. |
|---|---|---|---|---|---|---|---|---|---|---|---|---|---|---|---|---|---|---|---|---|---|---|---|---|
| Video-LLaMA (70) | CLIP-G | 7B | 27.5 | 25.5 | 51.0 | 29.0 | 39.0 | 48.0 | 40.5 | 38.0 | 22.5 | 22.5 | 43.0 | 34.0 | 22.5 | 32.5 | 45.5 | 32.5 | 40.0 | 30.0 | 21.0 | 37.0 | 34.1 |
| LLaMA-Adapter (71) | ViT-B | 7B | 23.0 | 28.0 | 51.0 | 30.0 | 33.0 | 53.5 | 32.5 | 33.5 | 25.5 | 21.5 | 30.5 | 29.0 | 22.5 | 41.5 | 39.5 | 25.0 | 31.5 | 22.5 | 28.0 | 32.0 | 31.7 |
| Video-ChatGPT (40) | ViT-L | 7B | 23.5 | 26.0 | 62.0 | 22.5 | 26.5 | 54.0 | 28.0 | 40.0 | 23.0 | 20.0 | 31.0 | 30.5 | 25.5 | 39.5 | 48.5 | 29.0 | 33.0 | 29.5 | 26.0 | 35.5 | 32.7 |
| VideoChat (24) | CLIP-G | 7B | 33.5 | 26.5 | 56.0 | 33.5 | 40.5 | 53.0 | 40.5 | 30.0 | 25.5 | 27.0 | 48.5 | 35.0 | 20.5 | 42.5 | 46.0 | 26.5 | 41.0 | 23.5 | 23.5 | 36.0 | 35.5 |
| VideoChat2 (25) | UMT-L | 7B | 66.0 | 47.5 | 83.5 | **49.5** | 60.0 | 58.0 | **71.5** | **42.5** | 23.0 | 23.0 | 88.5 | 39.0 | 42.0 | 58.5 | 44.0 | 49.0 | 36.5 | 35.0 | 40.5 | **65.5** | 51.1 |
| ST-LLM (37) | BLIP2 | 7B | 66.0 | 53.5 | **84.0** | 44.0 | 58.5 | **80.5** | 73.5 | 38.5 | 42.5 | 31.0 | 86.5 | 36.5 | 56.5 | 78.5 | 43.0 | 44.5 | 46.5 | 34.5 | 41.5 | 58.5 | 54.9 |
| GPT-4V | Unk | GPT-4 | 55.5 | **63.5** | 72.0 | 46.5 | 73.5 | 18.5 | 59.0 | 29.5 | 12.0 | 40.5 | 83.5 | 39.0 | 12.0 | 22.5 | 45.0 | 47.5 | 52.0 | 31.0 | 59.0 | 11.0 | 43.5 |
| PLLaVA 7B | ViT-L | 7B | 58.0 | 49.0 | 55.5 | 41.0 | 61.0 | 56.0 | 61.0 | 36.0 | 23.5 | 26.0 | 82.0 | 39.5 | 42.0 | 52.0 | 45.0 | 42.0 | 53.5 | 30.5 | 48.0 | 31.0 | 46.6 |
| PLLaVA 13B | ViT-L | 13B | 66.0 | 53.0 | 65.5 | 45.0 | 65.0 | 58.0 | 64.5 | 35.5 | 23.5 | 30.0 | 85.0 | 39.5 | 45.5 | 57.0 | 47.5 | 49.5 | 49.0 | 33.0 | 53.0 | 37.0 | 50.1 |
| PLLaVA 34B | ViT-L | 34B | **67.5** | 53.0 | 82.0 | 47.0 | **79.0** | 68.5 | 67.5 | 36.5 | **37.5** | **49.5** | **91.0** | 40.5 | **43.0** | 70.0 | 51.5 | 50.0 | 66.5 | 39.5 | 63.5 | 59.0 | 58.1 |
| Improve over GPT-4V | - | - | 12.0 | -10.5 | 10.0 | 1.5 | 5.5 | 50 | 8.5 | 7.0 | 25.5 | 9.0 | 7.5 | 1.5 | 31.0 | 57.5 | 5.5 | 2.5 | 14.5 | 8.5 | 4.5 | 48.0 | 14.5 |

Table 3: Results on MVBench multi-choice question answering.

Table 2 demonstrates the results on VideoQA. PLLaVA 34B significantly outperforms all the existing methods on the Accuracy and Score metrics of MSVD, MSRVTT, ActivityNet, and TGIF. Compared to GPT-4V, PLLaVA 34B achieves improvement margins of 3.6, 4.9, 3.9, and 15.3 on these four benchmarks. The performance of PLLaVA with 7B and 13B model sizes also exceeds all the baselines on the Score metric. These results not only prove the capability of our model in conducting video question answering but also highlight the superiority of our pooling strategy in scaling model size.

PLLaVA also outperforms baselines in the average VCG score. The 7B, 13B, and 34B versions have all outperformed their best counterparts of the same LLM size, with margins of 2.9%, 7.1%, and 12.6%, respectively. Notably, PLLaVA achieves superior performance on CI(correctness of information), DO(Detail Orientation), and CU(Context Understanding) compared to the previous SOTA, with 34B exceeding them by 5.8%, 6.7%, 9.2%. These results indicate that PLLaVA will be of great potential to do detailed video captioning. As for TU(temporal understanding), PLLaVA 34B exceeds its fair opponent IG-VLM LLaVA 34B by 6%. Compared with models that utilize the specialized video encoder, VideoChat2, or a more complicated frame combination method, Chat-Univ, PLLaVA still has some room for improvement by fingering the pooling strategy or incorporating a better vision encoder. CO(Consistency) measures generation consistency when the model encounters different questions that lead to similar answers. Compared to baselines except for IG-VLM, our model achieves much better consistency.

MVBench is a comprehensive video understanding benchmark, focusing on questions that require overall comprehension of multiple frames. As shown in Table 3, PLLaVA surpasses the previous SOTA VideoChat2 with a margin of 13.7% on average across 20 tasks. If we look into each aspect of MVBench, our method performs very well, concerning 17 out of 20 tasks of MVBench, which shows that our model has the superiority to understand many fine-grained details about videos accurately.

However, we also noticed some aspects of our model still need to improve, such as CI(CounterFactual Inference) and OS(object shuffle). CI is used to predict what might happen if an event occurs, and OS is used to locate the final position of an object in an occlusion game. These two require strong reasoning ability and imagination to answer. VideoChat2 is pretrained with a large amount of video data with a specialized video encoder and fine-tuned with both video and image reasoning data, thus presenting better performance in these aspects.

We also present the results of PLLaVA on two recent benchmarks: VideoMME (9), a comprehensive dataset with varying video lengths and high-quality annotations, and LongVideoBench (57), designed specifically for long video understanding. We compare PLLaVA with

| Method | VideoMME | LongVideoBench | VideoQA |
|---|---|---|---|
| VILA 40B | 63.2 | - | 64.8 |
| Gemini 1.5 Pro | 75.0 | 52.7(16frame) | - |
| LLaVA-Next-Video 34b | 52.0 | 50.5 | - |
| PLLaVA 34b | 54.0 | **53.2** | **72.5** |

Table 4: Results on VideoMME, LongVideoBench and average VideoQA score in Table 2.

its most similar counterpart, LLaVA-Next-Video, which utilizes the same backbone models and applies a pooling strategy during training. The results demonstrate that PLLaVA outperforms LLaVA-Next-Video in both standard and long-video comprehension tasks. Additionally, we compare PLLaVA to the recent proprietary model Gemini 1.5 Pro and the VILA model (31), which employs a well-trained video encoder and is fully trained on extensive image and video datasets. When using the same number of frames, PLLaVA achieves results comparable to Gemini 1.5 Pro. In the VideoMME, PLLaVA produces decent results, despite not undergoing full LLM training or utilizing a specialized video encoder. For VideoQA, PLLaVA outperforms VILA.

## 4.4 ANALYSIS

Our PLLaVA is a simple and parameter-efficient method to adapt image MLLMs into the video domain. We also provide a feasible way to scale the models to larger sizes, which we found is hard to achieve in other methods such as ChatUniv (18) and IG-VLM (20). In the following, we further provide some analysis related to the explanations on pooling shapes and the influence of LoRA weight on different tasks.

**Image? Video? or Both?** Post-training optimization is defined as the combination of the LLMs' parameters of image MLLMs and learned LLMs' LoRA weights from video samples. A suitable fusion ratio could be highly efficient in boosting model performance trained under low-quality video-text samples. Here, we discuss the influence of different choices of fusion ratio on the understanding performance. As shown in Figure 6, the x-axis represents the alpha

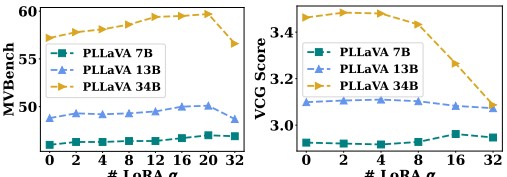

Figure 6: Post Optimization Effects with LoRA $\alpha$.

value of LoRA. 0 indicates no LoRA weights added, and 32 means the LoRA weights are fully applied to LLMs. We observed distinct trends between MVBench and VCG Score. The former exhibits a peak around alpha 20, while the latter performs best near alpha 4. This variance can be attributed to the nature of these two benchmarks: VCG typically involves longer length generations, whereas MVBench focuses on multiple-choice question answering, placing less emphasis on language generation ability. Consequently, weights learned from video-text data samples are more tailored for MVBench tasks. In this way, a larger portion of video weights are beneficial for MVBench. Moreover, from these two figures, it's evident that combining video and image weights leads to better performance than at the extremes of 0 and 32.

## 5 CONCLUSION

In this paper, we conduct an initial investigation for extending image-language models to videos with a simple yet extremely effective method, termed PLLaVA . With the new model, it is easier to scale the training with more data and larger large language models with a more controllable strategy for over-training and performance saturation. PLLaVA 's ability to give detailed captions also contributes to the community development of multimodal understanding and generation.

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

## A    MORE RELATED WORKS

**Pipeline-based Video Understanding.** By encompassing various video foundation models and prompting techniques on LLMs, pipeline-based video understanding has extensively explored for video captioning and question-answering (68; 53; 5; 69; 3; 24; 40). Typically, this approach involves converting videos into textual elements with models such as event localization, objection detection, and image captioning, which are then integrated with an LLM in the final phase. By representing videos as text tokens, it harnesses the LLMs' proficiency in processing textual data, thereby permitting the interpretation of temporal sequences via these crafted descriptions.

**Video-Text Pretraining.** Another track of work focuses on pretraining foundation models on large-scale video-text datasets (72; 63; 6; 66; 56; 55), which could be used in downstream video understanding tasks. LaViLa (72) employs smaller LLMs, e.g. T5 (44) and GPT-2 (43), to deal with visual features. Vid2Seq further enhances the video pretraining on T5 with fine-grained video captions and the time token technique to focus on event boundaries. VAST (6) advances video-text retrieval with multiple modality inputs. Merlin (66) follows the training pipelines of image domain MLLMs (34; 23) and introduces the foresight training technique specialized for video on much larger LLMs like Vicuna 7b (73). However, these methods highly demand computing resources and are usually not designed for general-purpose video understanding tasks.

**Video Input Compression.** To deal with long video input, MovieChat (47) implemented a novel memory-based mechanism within transformers, combining similar frames to reduce both computational load and memory footprint. Chat-UniVi (18) debuted a harmonized approach for processing images and videos, condensing spatial and temporal tokens through dynamic token merging. LLaMA-VID (27) innovates with a dual-token approach, allowing for more efficient compression.

**Full-trained Video LLMs.** Another avenue of research (25; 52; 31; 7; 22; 35; 38) requires substantially more computational resources and training data. These studies typically utilize image or video foundation encoders (26; 54; 55) with Large language models and engage in both pre-training and instructional tuning to develop video-aware large language models (LLMs). While these approaches generally demonstrate significantly stronger video comprehension capabilities, they also incur higher costs. Our work mainly focuses on problems of parameter-efficient adaptation form image MLLMs to video domain.

## B    EXTRA QUATITATIVE EXPERIMENTS

We introduce results on some newly proposed benchmarks, including VideoMME and LongVideoBench. PLLaVA shows better results than LLaVA-Next-Video and competitive results with Gemini 1.5 Pro if using the same number of frames.

| Method | VideoMME | LongVideoBench |
|---|---|---|
| VideoChat2 Mistral 7b | 39.5 | 43.5 |
| LLaVA-Next-Video 34b | 52.0 | 50.5 |
| Gemini 1.5 Pro | 75.0 | 52.7(16frame) |
| PLLaVA 7b | 42.8 | 40.2 |
| PLLaVA 13b | 47.2 | 45.6 |
| PLLaVA 34b | 54.0 | 53.2 |

Table 5: Results on VideoMME and LongVideoBench.

## C    HUMAN EVALUATION

We have also presented preliminary human evaluation results in Table 6. We randomly selected 20 samples that were doing detailed caption tasks and asked three individuals to evaluate each sample. The results were compared across three aspects: correctness, completeness, and detail. The results demonstrate that our PLLaVA significantly outperforms IGVLM in all three aspects from the human evaluators' perspective when doing dense captioning for videos.

| Metrics | Win | Tie | Lose |
|---|---|---|---|
| Correctness | 45 | 52.5 | 2.5 |
| Completeness | 45 | 55 | 0 |
| Detail | 57.5 | 30 | 12.5 |

Table 6: Human evaluation result by comparing PLLaVA 34b vs. IG-VLM 34b.

## D ANALYSIS

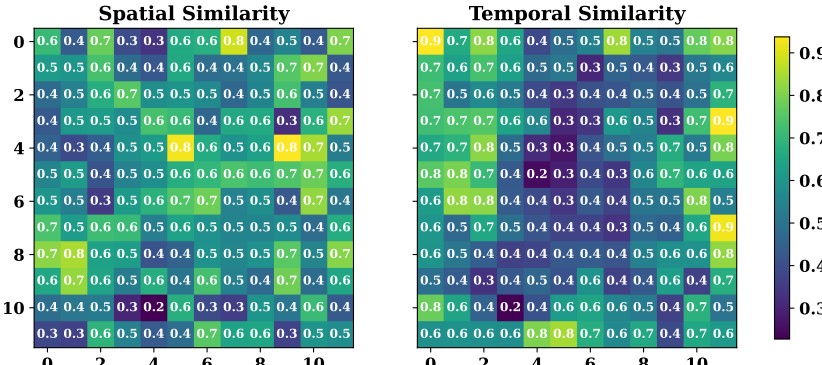

Figure 7: Vision token embedding similarities between spatial token neighbors and temporal token neighbors.

**Temporal or spatial pooling?** In Sec.4.2, we have illustrated the impact of temporal and spatial poolings, concluding that pooling along the temporal dimension consistently results in decreased performance compared to retaining the original frame numbers. We attribute this phenomenon to the interference with token features. In image MLLMs, features are derived from images/video frames using CLiP-ViT models, which produce embedded patches for each image/video frame, resulting in a video feature with shape $(T, H, W)$. Pooling changes the dimensions of $T$ (time), $H$ (height), and $W$ (weight). In contrast to pooling along the spatial dimension (local pooling on single images/frames, changing $H$ and $W$), pooling along the temporal dimension (changing $T$) risks altering the original frame features. To validate the guess, we visualize token similarities among spatial and temporal token neighbors for a video feature in Figure 7. The two subfigures reveal significantly higher similarities within spatial neighbors than temporal neighbors. This observation supports the potential distortion of original token features caused by temporal pooling. LLMs are designed for sequence understanding. Even without preprocessing on temporal information aggregation, they can model temporal relations.

## E CASE STUDIES

Apart from these quantitative results, we also qualitatively investigate the video understanding abilities of PLLaVA models. We have shown several caption examples in Figure 8. According to the video clips, compared to IG-VLM, PLLaVA 34B recognizes more details about videos, including the clothes worn by the main characters, the environment, and even some of the words in this video. Besides, PLLaVA can better understand the video content, in which people are playing badminton rather than volleyball. These mistakes made by IG-VLM could be caused by the lowered resolution when concatenating frames into the grid view in the method design. Pooling reduces dimension after frames are encoded, thus leading to less information loss.

## F DENSE RECAPTION

In view of the caption ability of PLLaVA , we further tested its recaption task and contributed 1K video Inter4K (49) caption dataset. An example is shown in Figure 9. Compared to Open-Sora

What happens in the video?

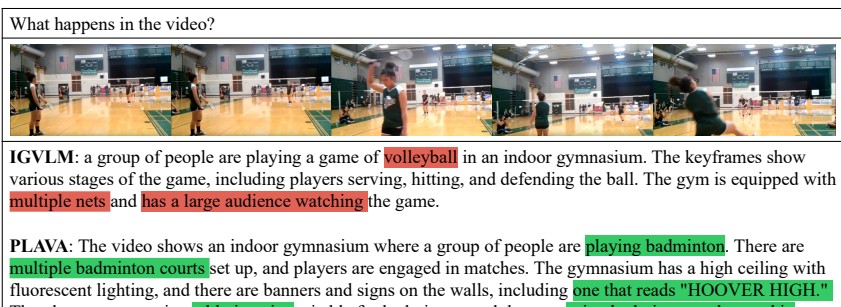

**IG-VLM 34b**: a man is playing a saxophone in various indoor settings, likely a shopping mall or a public space, with different lighting and background elements.

**PLAVA 34b**: In the video, there is a man playing a saxophone in what appears to be a public space, possibly a mall or a street. He is dressed in a suit and tie, which suggests that he might be a professional musician performing for an audience. There are other people in the background, indicating that this is a public area with some foot traffic. The setting looks like it could be a shopping center or a similar venue where live music is sometimes performed to entertain visitors.

(a) Street Saxophone.

What happens in the video?

**IGVLM**: a group of people are playing a game of volleyball in an indoor gymnasium. The keyframes show various stages of the game, including players serving, hitting, and defending the ball. The gym is equipped with multiple nets and has a large audience watching the game.

**PLAVA**: The video shows an indoor gymnasium where a group of people are playing badminton. There are multiple badminton courts set up, and players are engaged in matches. The gymnasium has a high ceiling with fluorescent lighting, and there are banners and signs on the walls, including one that reads "HOOVER HIGH." The players are wearing athletic attire suitable for badminton, and they are using badminton rackets to hit shuttlecocks back and forth across the net

(b) Badminton match.

Figure 8: Case Studies.

GPT-4 pipeline, our model captures better caption details and also highlights motion information in the video, demonstrate PLLaVA 's potential to contribute to the video generation community.

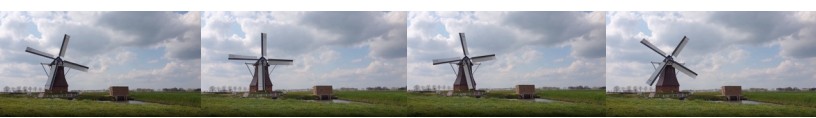

PLLaVA 34B

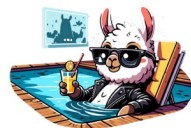

The video shows a large, traditional Dutch windmill situated on a grassy area next to a body of water, possibly a river or a canal. The windmill has a prominent presence in the scene, with its sails visible and appearing to be in motion. In the background, there is a small building, which could be a storage shed or a house, and a fence. The sky is partly cloudy, suggesting a changeable weather condition. There are no characters or actions visible in the video, as it is a static scene with no people or animals present. The focus of the video is on the windmill and its surroundings, providing a glimpse into the traditional landscape of the Netherlands.

Open-Sora GPT-4
Caption Pipeline

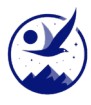

The video showcases a traditional Dutch windmill set against a partially cloudy sky, with a gently flowing waterway in the foreground. Surrounding the windmill is a lush green field dotted with yellow wildflowers. A small outbuilding is adjacent to the windmill, separated by a quaint wooden fence. The peaceful rural landscape suggests the tranquil pace of life in the count side, with the windmill standing as an iconic symbol of Dutch heritage. The windmill's sails are stationary, implying a calm day without much wind.

Figure 9: Recaption comparison between PLLaVA 34B and Open-Sora.

