# OpenReview forum: "PLLaVA: Parameter-efficient LLaVA Extension from Image to Video Understanding"
_ICLR.cc/2025/Conference — Submitted to ICLR 2025_

### Official Review · Reviewer_EneY · 2024-10-21

**Soundness:** 2
**Presentation:** 2
**Contribution:** 2
**Rating:** 5
**Confidence:** 5

**Summary:**

The proposed approach PLLaVA adapts image vision-language models (VLMs) for video understanding tasks in a parameter-efficient way. The authors introduce an adaptive pooling strategy that smooths out feature distributions across the temporal dimension. This method termed Pooling LLaVA, prevents issues like dominant feature tokens, which can degrade model performance and leads to improvements in video-language tasks such as video question answering (QA) and video captioning. The authors test PLLaVA’s performance compared to state-of-the-art models across multiple benchmarks, including MVBench, VideoMME and Video-ChatGPT.

**Strengths:**

* The approach allows image-based VLMs to be adapted for video tasks using adaptive pooling. This avoids computation costs associated with training entirely new models for video data.
* The analysis of pooling applied to spatial vs temporal dimensions (Sec. 4.2) is interesting.
* The analysis demonstrating the issue of dominant tokens (high-norm visual feature tokens that skew performance during model fine-tuning) is novel and provides new insight for VLMs. However, the pooling approach concluded from this analysis is not new.
* PLLaVA is tested on a wide range of video-related tasks and benchmarks. The approach is reported to perform reasonably well overall.

**Weaknesses:**

* The novelty seems to be limited in the proposed approach since the main contribution is spatial pooling instead of temporal pooling and the application of LoRA weight aggregation.
* The main source of improvement seems to be coming from fine-tuning the LLaVA-Next image model on videos. What are the technical improvements over the LLaVA-Next model?
* Within the same parametric complexity models (e.g., 7B), the PPLaVA approach performs lower than the competing models like IG-VLM.
* It is unclear how the model handles complex temporal dynamics inherent in video tasks. E.g., tasks that involve intricate motion understanding or multi-event reasoning. Can the authors provide specific examples of how PLLaVA handles long temporal dependencies in complex video scenes?
* Although the empirical results show that pooling helps alleviate the problem of dominant tokens, the theoretical explanation behind why spatial pooling works is somewhat lacking. The provided justification is generic for spatial and temporal pooling, and the explaination on softmax distributions and sharp/balanced token distributions seems quite generic, not justifying why specifically spatial pooling is better.
* The writing needs to be significantly improved. Several details are not clear. There are some typos as well: e.g., Fig. 4 "Adaptative" --> "Adaptive".

**Questions:**

* Eq. 2, AdaptStructPooling is not properly defined. How is this pooling function implemented, how is it different from the pooling operations implemented in VideoChatGPT (I understand the only difference seems to be pooling on the spatial dimension)? Isn't the problem of dominant high-norm visual tokens already addressed by VideoChatGPT with their average pooling?
* Section 3.4, it is unclear how the Post Optimization approach differs from LoRA? Its also similar to distillation where a model weights are used to guide the adapted model weights (e.g., in continual learning). This needs to be clarified and clearly explain.
* Can you elaborate on how PLLaVA’s pooling strategy impacts its ability to handle long-term temporal reasoning?Using the tests on tasks involving longer videos, kindly explain how did the performance compare to short clips?
* The pooling strategy is designed to alleviate the issue of dominant tokens, but how does it affect specific visual features? For example, does the pooling lead to a loss of finer details in complex scenes with many objects or fast-moving elements?
* Table 4, authors compare PLLaVA with its most similar counterpart, LLaVA-Next-Video, however its not clear specifically what changes are leading to the improvement over LLaVA-Next-Video? An ablation on this model will help clarify.

---

> ### Author Response · Authors · 2024-11-24
> **Novelty and Contributions, Pooling details**
>
> > The novelty seems to be limited in the proposed approach since the main contribution is spatial pooling instead of temporal pooling and the application of LoRA weight aggregation.
>
> Our contribution is: `By pinpointing the possible causes of the data and model scaling diffculties, we propose an efficient and effective method to build video understanding models from image MLLMs.` PLLaVA shows that pooling along the spatial dimension and learning temporal information with LLMs is already effective for video understanding.`No fancy technique in temporal processing does not indicate the limitation in novelty`. As for `LoRA weight aggregation`, it is also a new simple but very effective method that avoids the degradation of LLMs during video-text training.
>
> ---
>
> > The main source of improvement seems to be coming from fine-tuning the LLaVA-Next image model on videos. What are the technical improvements over the LLaVA-Next model?
>
> Our contribution, as outlined in the previous discussion, is to propose an efficient approach for developing video understanding models using image-based MLLMs, such as the LLaVA-Next image model. Specifically, `we introduce an effective pooling strategy and post-optimization techniques to facilitate data and model scaling when training video LLMs with diverse quality video-text data`.
>
> ---
>
> >It is unclear how the model handles complex temporal dynamics inherent in video tasks. E.g., tasks that involve intricate motion understanding or multi-event reasoning. Can the authors provide specific examples of how PLLaVA handles long temporal dependencies in complex video scenes?
>
> For a video input, PLLaVA uniformly choose N frames, then we use the AdaptStructurePooling to reduce the dimesions of the visual input. The visual input are tranformed into token into the LLM. The LLM model then learns temporal information through the LLMs sequential modeling ability. This operation fully utilize the visual understanding
>
> ---
>
> >Although the empirical results show that pooling helps alleviate the problem of dominant tokens, the theoretical explanation behind why spatial pooling works is somewhat lacking. The provided justification is generic for spatial and temporal pooling, and the explaination on softmax distributions and sharp/balanced token distributions seems quite generic, not justifying why specifically spatial pooling is better.
>
> Thanks the reviewer's comments but there might be a misunderstanding. `In Lines 313-340, we show pooling operation are effective in alleviating dominant tokens because its smoothes the input distributions. This applies both to temporal and spatial pooling. Therefore, there is no need to justify for spatial again.`The reviewer may comprehend as only the combination of temporal and spatial works.
>
> ---
>
> > Eq. 2, AdaptStructPooling is not properly defined. How is this pooling function implemented, how is it different from the pooling operations implemented in VideoChatGPT (I understand the only difference seems to be pooling on the spatial dimension)?
>
> Thank you for pointing that out, we will add more details about the term AdaptStructurePooling. AdaptStructurePooling refers to the [AdaptiveAvgPool3d](https://pytorch.org/docs/stable/generated/torch.nn.AdaptiveAvgPool3d.html#torch.nn.AdaptiveAvgPool3d) layer defined in pytorch. `It can pool the input video feature $X\in R^{T\times h \times w\times d}$ into any specified size, thus is capable of pooling in both the temporal and spatial dimensions`. Since it is a straightforward and commonly used layer, we did not provide a detailed definition initially.
>
> `In VideoChatGPT, the input is averaged from video features in both spatial and temporal dimensions.` In specific, given the input video feature $X\in R^{T\times h \times w\times d }$, VideoChatGPT gets the spatial feature $X_s \in R^{ h \times w\times d }$ by pooling along the temporal dimension, and the temporal feature $X_t \in R^{ T\times d }$ along the spatial dimension. The visual input is then defined as the concatenation $X_v := (X_s,X_t)\in R^{ (T+h\times w) \times d }$. `This brutal average(pooling) operation could destroy the detailed information in each video frame due to the big strides.`
>
> ---
> > Isn't the problem of dominant high-norm visual tokens already addressed by VideoChatGPT with their average pooling?
>
> We want to remind the reviewer that `Dominant token is what we found during our attempts to adapt image MLLM into video domain with raw video frame features. VideoChatGPT's ineffective in video understanding lies in its brutal manipulation on the visual features, which cause a lot of information loss for each frame, limits the upper bound of video understanding`. This further validates the necessity of a proper pooling strategy to both reduce training cost and improve effectiveness.

---

> ### Author Response · Authors · 2024-11-24
> **PLLaVA Pooling Stratgy Effects.**
>
> > Can you elaborate on how PLLaVA’s pooling strategy impacts its ability to handle long-term temporal reasoning?Using the tests on tasks involving longer videos, kindly explain how did the performance compare to short clips?
>
> Our experimental findings, presented in Table 4, highlight the performance of PLLaVA in the LongVideoBench. PLLaVA achieves a score of 53.2, outperforming both LLaVA-Next-Video 34b and Gemini 1.5 Pro (52.7 with 16 frames), `demonstrating its superior capability in handling long videos`. Furthermore, detailed analysis of PLLaVA's performance in VideoMME indicates that it consistently delivers strong results across a range of video lengths when compared to LLaVA-Next-Video.
>
> |Method|Short|Medium|Long|
> |---|---|---|---|
> |LLaVA-Next-Video 34B|61.2|50.1|44.3|
> |PLLaVA 34B|64.1|51.6|46.0|
>
> ---
>
> > The pooling strategy is designed to alleviate the issue of dominant tokens, but how does it affect specific visual features? For example, does the pooling lead to a loss of finer details in complex scenes with many objects or fast-moving elements?
>
> Thank the reviewer's good question. PLLaVA use stride 2 to pool the spatial dimension, which makes slight effects on the visual features.
> PLLaVA presents superior ability of giving detailed video descriptions, since it achieve very good performance in the benchmark Video-ChatGPT, as shown in Table 2. Furthermore, PLLaVA also show good performance when dealing with fast moving elements. For example, PLLaVA 34B also achieve best results for detecting the Moving Attribute in the MVBench, as shown in Table 3.

---

> ### Author Response · Authors · 2024-11-24
> **Post Optimization and other experiments**
>
> > Section 3.4, it is unclear how the Post Optimization approach differs from LoRA? Its also similar to distillation where a model weights are used to guide the adapted model weights (e.g., in continual learning). This needs to be clarified and clearly explain.
>
> Post Optimization is a post processing after finetuning the MLLM model with video-data. Post Optimization adjusts the ratio of adding the learned LoRA weights to the original model. It remains more of the original LLM weights to keep the languague ability. Distillation is a technique that uses teacher weights to guide the training of the student's model. It requires finetuning while Post Optimization only adjust weights without training.
>
> ---
>
> > Table 4, authors compare PLLaVA with its most similar counterpart, LLaVA-Next-Video, however its not clear specifically what changes are leading to the improvement over LLaVA-Next-Video? An ablation on this model will help clarify.
>
> The differences between LLaVA-Next-Video and PLLaVA are as follows:
> - Both PLLaVA and LLaVA-Next-Video use pooling strategy. PLLaVA furtehr utilizes the post-optimization method to calibrate the model parameters.
> - PLLaVA only trains with LoRA parameter while LLaVA-Next-Video trains all LLM parameters.
> - From the training strategy, the resource of the 860K data used in LLaVA-Next-Video is not known. We use the 783k public video-text data.
> With fewer training parameters and data, we could get better results.
>
> >Within the same parametric complexity models (e.g., 7B), the PPLaVA approach performs lower than the competing models like IG-VLM.
>
> In Table 2, we have shown `PLLaVA 7B outperforms IG-VLM 7B in most metrics(10 out of 14 columns)` with very high margins. Therefore it is NOT true that PLLaVA performs lower.

---

> > ### Comment · Reviewer_EneY · 2024-11-27
> >
> > I want to thank the authors for their responses. I appreciate their effort, but the approach has a number of heuristics and it am still not satisfied with the novelty and clarity. The requested ablation would have been helpful to clarify the source of improvements. I will keep my score.

---

> > > ### Author Response · Authors · 2024-11-27
> > > **Clarification on the improvement over LLaVA-Next-Video**
> > >
> > > **We appreciate the reviewer's feedback and explanations. However, we kindly request that the reviewer reconsider our contributions and the necessity of the ablation experiment, and to potentially reevaluate the decision regarding the score.**
> > >
> > > **Novelty**: `Our research provides a novel analysis on what could affect the adaptation from image MLLMs to the video domain, the issue of high-norm tokens and Low-quality data.`
> > > To address these challenges, we propose PLlaVA, a solution that effectively tackles both aspects. The pooling strategy is designed to achieve a smooth and balanced visual distribution while introducing more frames, thereby eliminating the high-norm visual token issue and facilitating data scaling. Additionally, a novel post-optimization method is employed to preserve the language capabilities of the LLM, which was harmed when training with low-quality data.
> > >
> > > **Ablation Study**: As illustrated in Figure 3(c), `the ablation study already demonstrates the significant effectiveness of post-optimization, particularly with the 34B model`.  We believe that a direct comparison with LLaVA-Next-Video is unnecessary, as these two models employ distinct architectures and are trained on different datasets.

---

### Official Review · Reviewer_vomH · 2024-10-29

**Soundness:** 3
**Presentation:** 3
**Contribution:** 3
**Rating:** 6
**Confidence:** 4

**Summary:**

This work focuses on video undersdtanding. It analyzed current drawback for Vision-Language Model on video training and proposed to extend Vision-Language Model to support video with pooling operations. Experiments prove effectiveness on several video datasets.

**Strengths:**

1. This work proposes to extend current Vision-Language Model to support video, which is important and promising.
2. Extensive experiments on video-based benchmarks prove the effectiveness.
3. The overall method is simple and easy to follow.

**Weaknesses:**

It is interesting to find training with pooling can help video training. However, there could be some concerns regarding this paradigm that need to be solved.

1. Whether this phenomenon is caused by the low-quality video training data? Because most of the visual tokens in video dataset could be redundant for simple video caption. A suggestion is to use high-quality data for model training, like LLaVA-Video [A] data.

2. Whether the pooling strategy harms the performance on image? It's good to find the authors counduct extensive experiments for video. However, the performance on image-based settings is still not clear. Considering add image-based experiments to make it more clear.

3. What's the baseline used in the experiments? Considering add experiments over the baseline to make the improvement more clear.

4. LLaVA-Next-Video 34b in Table 4 should also be added in Table 2 and 3.

[A] Video Instruction Tuning With Synthetic Data, 2024

**Questions:**

More studied are recommended to futher varified the proposed strategy. Please refer to the questions on the Weakness section.

---

> ### Author Response · Authors · 2024-11-26
>
> ### Phenomenon with low-quality of data
>
> >Whether this phenomenon is caused by the low-quality video training data? Because most of the visual tokens in video dataset could be redundant for simple video caption. A suggestion is to use high-quality data for model training, like LLaVA-Video data.
>
> Yes, low quality data should be the cause of the model size scaling failure. PLLaVA targets to build video understanding without change the data quality. We will consider include LLaVA-Video data in the future extension of PLLaVA.
>
> ---
> ### Add more results of LLaVA-Video-Next
> >What's the baseline used in the experiments? Considering add experiments over the baseline to make the improvement more clear. LLaVA-Next-Video 34b in Table 4 should also be added in Table 2 and 3.
>
> Thank you for pointing that out. LLaVA-Next-Video uses a different OpenAI model(instead GPT-3.5-turbo-0125 used by most of the baselines) when evaluating on VCG, so we cannot add it into Table 2 for fair comparison. For Table 3,LLaVA-Next-Video didn't test in MVBench. In the following table we provide we show that PLLaVA significantly outperform LLaVA-Next-Video in the benchmark ActivityQA. We will add this to Table 2 in the revised version.
>
> |Method|ActivityQA(Acc/Score)|
> |--|--|
> |LLaVA-Next-Video 34B|58.8/3.4|
> |PLLaVA 34B|60.9/3.7|
>
> ---
> ### Image Performance
> >Whether the pooling strategy harms the performance on image? It's good to find the authors counduct extensive experiments for video. However, the performance on image-based settings is still not clear. Considering add image-based experiments to make it more clear.
>
> The pooling we used in PLLaVA could negatively affect the performance of image since we didn't use image data during our training. We will consider add more evaluations on image benchmarks in the future.

---

### Official Review · Reviewer_ws3S · 2024-11-02

**Soundness:** 2
**Presentation:** 3
**Contribution:** 2
**Rating:** 5
**Confidence:** 4

**Summary:**

This paper proposes an efficient approach to adapting pre-trained image-language models for dense video understanding. The authors conducted several initial studies to analyze the limitations of applying image-based MLLMs directly to video. Initial experiments revealed that simply fine-tuning these models on video data could lead to performance issues. To address this, the authors propose a temporal pooling strategy that smooths feature distribution across time, reducing the impact of extreme feature values. They also introduce a post-training model merging technique that mitigates forgetting in large language models during multi-modal fine-tuning due to low-quality video-text data. PLLaVA demonstrates strong performance across various video benchmarks.

**Strengths:**

The paper is well-written, covering the motivation, methodology, and experiments. The authors first analyze the issues and challenges in directly applying image MLLMs to video understanding and then propose a simple pooling operation and post-training strategy to address these issues. They also provide extensive ablation studies to demonstrate the effectiveness of the design. Finally, this approach shows strong performance across various benchmarks.

**Weaknesses:**

1. Comparing PLLaVA with the 4-Frame method in the initial failure analysis does not seem fair if PLLaVA uses 16 frames, which includes more video content to inform the answer. Recent studies show that using more frames can significantly improve the results [1][2]
2. The author mentioned that adding 249K data from VideoChat2 to train Video-ChatGPT does not continue to improve the model; on the contrary, the performance worsens. However, this outcome may also be influenced by the quality or distribution of the 249K data, as recent papers have shown that data quality and data mixture strategies can significantly impact model performance. Showing the performance of Video-ChatGPT trained on this 249K data alone could help verify this conclusion.
3. Lack the explanation of the x/y axis for Figure 2(a) and is there a direct correlation between norm distribution and model performance?
4. The post-optimization steps seem a bit counterintuitive. Does this mean that fine-tuning makes the alignment between vision and text even worse? How does this compare with unfreezing the LLM during fine-tuning?
5. Missing a few baseline methods such as [1][2][3]

[1] Yuanhan Zhang, Bo Li, haotian Liu, Yong jae Lee, Liangke Gui, Di Fu, Jiashi Feng, Ziwei Liu, and Chunyuan Li. LLaVA-NeXT: A strong zero-shot video understanding model, 2024b. URL https://llava-vl.github.io/blog/2024-04-30-llava-next-video/.

[2] Xu, Mingze, et al. "Slowfast-llava: A strong training-free baseline for video large language models." arXiv preprint arXiv:2407.15841 (2024).

[3] Gao, Mingze, et al. "TC-LLaVA: Rethinking the Transfer from Image to Video Understanding with Temporal Considerations." arXiv preprint arXiv:2409.03206 (2024)

**Questions:**

1. Could you elaborate more on how the 4 frames are selected? It is possible that the performance worsens as more data samples are trained due to a mismatch between the selected frames and the text content. This could lead to irrelevant frames being selected, which in turn results in increased training and bigger misalignment between vision and text.
2. Is there any training involved in the post-optimization process? If so, what training data is used?
3. What is the possible reason for the performance drop when the spatial pooling shape is 20 and its increase back when the shape is 24 in Figure 5(a)?

---

> ### Author Response · Authors · 2024-11-26
>
> ### 1. Comparison between 4-Frame and PLLaVA
>
> > "Comparing PLLaVA with the 4-Frame method in the initial failure analysis does not seem fair if PLLaVA uses 16 frames."
>
> Due to the input length limitations of LLMs, `the maximum number of frames we can use in the N-frame method is 4`. PLLaVA with 16 frames actually has the same visual token length as the 4-frame method. From this perspective, it is fair. The only difference is that pooled visual tokens is better distributed.
> `Our failure analysis experiments is to demonstrate that inputting raw visual tokens into LLMs (N-frame) can encounter challenges related to data scaling, as shown in Figure 3(a) and 3(b).` Thus elicit the necessary design of pooling, which introduce better visual input distribution.
>
> > Could you elaborate more on how the 4 frames are selected? It is possible that the performance worsens as more data samples are trained due to a mismatch between the selected frames and the text content. This could lead to irrelevant frames being selected, which in turn results in increased training and bigger misalignment between vision and text.
>
> **We uniformly select 4 frames from videos just as we've done in PLLaVA**.
> We also agree that the sparsity information of chosen video frames could be another cause for the misalignments between videos and texts. Sparse frames also increase the sharpness of visual input distribution, thus more likely to lead to high-norm tokens. `Considering the constrain of the input length, we introduce pooling to both allow more frames and reduce the sharpness of visual distribution.`
>
> >What is the possible reason for the performance drop when the spatial pooling shape is 20 and its increase back when the shape is 24 in Figure 5(a)?
>
> AdaptiveStructurePooling layer is used for pooling. Given that the maximum size of the original feature is 24x24, transitioning to dimensions like 20x20 or 16x16 does not align with a natural pooling stride. This mismatch results in uneven pooling of the features, leading to unsuitable feature manipulation and potentially degrading performance.
>
> ### 2. Data Quality on VideoChatGPT
> > "The author mentioned that adding 249K data from VideoChat2 to train Video-ChatGPT does not continue to improve the model; on the contrary, the performance worsens. However, this outcome may also be influenced by the quality or distribution of the 249K data, as recent papers have shown that data quality and data mixture strategies can significantly impact model performance. Showing the performance of Video-ChatGPT trained on this 249K data alone could help verify this conclusion."
>
> Yes, data quality is crucial for the performance of trained models and here we agree VideoChatGPT was adversely affected by low-quality video-text data. `However, proper model structure and training strategy are essential for the resulted model trained on the same data(which is one of the motivation of our work)`. In our comparison, by using the same training data, we found that PLLaVA is more robust and scales effectively. This demonstrates that PLLaVA offers an excellent model architecture and training strategy.
>
> ### 3. Details of norm distribution in Figure 2(a)
> >Lack the explanation of the x/y axis for Figure 2(a) and is there a direct correlation between norm distribution and model performance?
>
> Thank you for bring that to our attention, we will add more details in the revised version. The y-axis represents the frequency of the norm values, while the x-axis displays the norm values themselves. When there are tokens with exceptionally high norms, such as those in the second and third rows of Figure 2(a), the generation performance can be adversely impacted.
>
> ### 4. Post optimization and Unfreeze LLM
> > The post-optimization steps seem a bit counterintuitive. Does this mean that fine-tuning makes the alignment between vision and text even worse? How does this compare with unfreezing the LLM during fine-tuning?
>
> Although it may seem counterintuitive, our findings are understandable. The reasoning is that training video LLMs with low-quality video-text data combined with LoRA weights degrades the LLM's language proficiency. Compared with LoRA, unfreezing the LLM requires a lot of GPU resources, which is why we did not pursue such experiments. In our comparison with LLaVA-Next-Video, a relevant ablation model that uses pooling and unfreezes LLM parameters during training, we observed that LLaVA-Next-Video 34B performs worse than PLLaVA 34B. `This highlights that the integration of LoRA with post-optimization not only enhances performance but also demands fewer computational resources than unfreezing LLM parameters.`
>
> >Is there any training involved in the post-optimization process? If so, what training data is used?
>
> No training is involved in the post-optimization.
>
> ### 5. Missing baselines
>
> For LLaVA-Next-Video[1], we have included it in Table 4. For Slowllava[2] and TC-LLaVA[3] we will add them to the revised version.

---

> ### Author Response · Authors · 2024-11-28
>
> Dear reviewer
>
> Thank the reviewer again for your comments. We're wondering whether our responses have resolved your questions or not?
>
> Best
> Authors of submission 4096

---

> > ### Comment · Reviewer_ws3S · 2024-12-02
> >
> > Thank the authors for the detailed responses. However, I still have concerns regarding the generalization and effectiveness of the proposed approach compared to using higher-quality SFT data or including video data during the pre-training stage, as the pooling strategy does not seem to specifically address the challenges of capturing useful temporal information or reducing redundant content in raw videos. Thus, I will keep my current score.

---

### Official Review · Reviewer_tQ9n · 2024-11-04

**Soundness:** 3
**Presentation:** 2
**Contribution:** 2
**Rating:** 5
**Confidence:** 4

**Summary:**

This paper presents PLLaVA, a parameter-efficient adaptation of LLaVA (Large Language and Vision Assistant) to extend its capabilities from image to video understanding. The authors highlight the challenges associated with directly fine-tuning image-language models on video data, such as performance degradation due to bias in visual features with high norms. To address these challenges, PLLaVA introduces an adaptive pooling strategy along the spatial and temporal dimensions to smooth feature distributions and reduce the influence of extreme feature values. This strategy aims to mitigate performance saturation commonly seen when directly using image models for video tasks. The authors also propose a post-training optimization technique to combine image-based and video-tuned model weights through weighted model merging of image-based VLM and video fine-tuned VLM. Experimental results show that PLLaVA achieves competitive or superior performance compared to existing models on several video-related benchmarks, including Video ChatGPT, MSVD-QA, MSRVTT-QA, MVBench, and ActivityNet-QA.

**Strengths:**

1. **Study of Feature Downsampling**: The authors provide a detailed study on the effect of feature downsampling in spatial and temporal dimensions, as well as model merging tuning. This could be valuable for practitioners who need to fine-tune related parameters for their specific applications.

2. **Post-Training Optimization**: The post-training optimization technique to mix weights from image and video domains is an interesting approach that helps retain image understanding while incorporating temporal dynamics. This allows PLLaVA to balance generalization capabilities between images and videos.

3. **Performance on Benchmarks**: The model shows measurable improvements over previous state-of-the-art models on multiple benchmarks. The improvements on MVBench are particularly notable, indicating PLLaVA's ability to better capture temporal dependencies, which is a key challenge in video-language understanding.

4. **Scalability**: The paper provides evidence that PLLaVA scales effectively across different model sizes (7B, 13B, 34B), maintaining or improving performance as parameters increase. This demonstrates its scalability without significant performance degradation.

**Weaknesses:**

1. **Presentation Issues**: The presentation of this work is difficult to follow, primarily due to several issues:

   1. **Concepts Without Explanation**: Several core concepts are introduced without sufficient explanation. For example, `AdaptStructurePooling`, which is crucial to the proposed approach, lacks a clear description of how it is implemented.
   2. **Logical Gaps**: There are several instances where claims are made without adequate examination. For instance, from L320 to L340, the authors mention that spiky input makes the output of the Softmax function less stable. However, there is no clear explanation of why this leads to tokens with high norms, which is claimed to be the effect of having inputs with spiky distributions. Such logical gaps hinder understanding of the motivation behind the proposed approach.
   3. **Confounding Conclusions**: In Section 4.2 on pooling layer design, the authors state that "pooling along the temporal dimension always downgrades model performance." However, in the following paragraph, they assert that "pooling over more video frames improves model efficiency and makes it more robust to user inquiries." This appears either conflicting or lacking context, making it unclear whether pooling is recommended or not.

2. **Technical Novelty**: Pooling or downsampling is already a widely adopted technique in feature engineering for visual language models. While the detailed parameter tuning is useful, the paper lacks a clear differentiation of how these results contribute to new techniques or novel insights.

3. **Complexity of Post-Training Optimization**: The process of tuning the mix ratio between image and video-trained weights is dataset-dependent, which could limit the practical adoption of the model. Providing more detailed guidance on selecting the mix ratio would make this approach more accessible to practitioners.

**Questions:**

1. **Post-Training Optimization Practicality**: Can the authors provide more detailed guidelines on how to effectively determine the mix ratio between image-based and video-trained weights during post-training optimization? This would help practitioners replicate the results without excessive trial and error.

2. **Data Quality Influence**: How does the quality of video-text datasets influence PLLaVA's performance? Would using higher-quality datasets or specific data augmentation strategies reduce issues related to dominant token norms and improve overall model performance.

---

> ### Author Response · Authors · 2024-11-24
> **Reply to presentation issues.**
>
> >"Concepts Without Explanation:...**AdaptStructurePooling**, which is crucial to the proposed approach, lacks a clear description of how it is implemented."
>
> Apologies for the misunderstanding, and thank you for bringing it to my attention. The term AdaptStructurePooling refers to the [AdaptiveAvgPool3d](https://pytorch.org/docs/stable/generated/torch.nn.AdaptiveAvgPool3d.html#torch.nn.AdaptiveAvgPool3d) layer defined in pytorch. It can pool the input video feature $X\in R^{T\times h \times w\times d}$ into any specified size, thus can do pooling in both the temporal and spatial dimensions. Since it is a straightforward and commonly used layer, we did not provide a detailed definition initially.
>
> ---
>
> >"Logical Gaps:.. from L320 to L340, the authors mention that spiky input makes the output of the Softmax function less stable. However, there is no clear explanation of why this leads to tokens with high norms, which is claimed to be the effect of having inputs with spiky distributions."
>
> Thanks for pointing that out, here we give a more detailed explanation.
> From L320 to L340, we show that Larger outputs (due to significant input features) cause substantial gradients. The optimizer adjusts these weights more prominently compared to others. Eventually, these enlarged weight cause part the of learned feature larger, thus leading to dominat token embeddings. We willl add a more detailed proof in the revised version.
>
> ---
>
> >"Confounding Conclusions:pooling along the temporal dimension always downgrades model performance." However, in the following paragraph, they assert that "pooling over more video frames improves model efficiency and makes it more robust to user inquiries." "
>
> Thank you for bringing this to our attention. We apologize for any misunderstandings it may have caused. In the paragraph beginning on line 422, we intend to explain that pooling is more robust in handling user inquiries. `The phrase "more video frames" is meant to compare this approach with the 4-frame method, rather than implying that pooling has been done over the temporal dimension.`

---

> ### Author Response · Authors · 2024-11-24
>
> ## Clarification on Techinical Novelty
>
> > Technical Novelty: Pooling or downsampling is already a widely adopted technique in feature engineering for visual language models. While the detailed parameter tuning is useful, the paper lacks a clear differentiation of how these results contribute to new techniques or novel insights.
>
> Our contribution is: `By pinpointing the possible causes of the data and model scaling diffculties, we propose an efficient and effective method to build video understanding models from image MLLMs.` Pooling, as a straightforward yet powerful technique, helps eliminate dominant tokens, thereby contributing to a stable training process. `The key insight of PLLaVA is that adapting image MLLMs for the video domain is both a practical and cost-effective strategy. By avoiding complex manipulations on video features and instead entrusting temporal learning to LLMs, we provide an elegant solution for developing video understanding models.`
>
> ---
>
> ## Clarification on of Post-Training Optimization and Data Quality
>
> > The process of tuning the mix ratio between image and video-trained weights is dataset-dependent, which could limit the practical adoption of the model. Providing more detailed guidance on selecting the mix ratio would make this approach more accessible to practitioners.
>
> Thank the reviewer's constructive suggestion. Post Optimization is to adjust the mix ratio of the original weight and the learned LoRA weight. Since the process requires no training, it is easy to do some grid search on the validation data to decide the ratio.  We also provide the effects and analysis of mix ratios in Figure 6(lines515-531). There are two criteria that can be concluded:
> - For tasks requiring detailed text generation, the mix ratio should be kept very low. In contrast, for tasks such as multiple-choice QA, the mix ratio $\alpha$ can be slightly lower than what is used during training.
> - Regarding model size, smaller models are less sensitive to the mix ratio $\alpha$, allowing for a range between 0 and the training ratio. However, larger models should adhere to the criterion specified above.
>
>
> > Data Quality Influence: How does the quality of video-text datasets influence PLLaVA's performance? Would using higher-quality datasets or specific data augmentation strategies reduce issues related to dominant token norms and improve overall model performance.
>
> Intuitively, video-text data with detailed text descriptions should be useful for the training of the model without lowering the language ability of the LLM.
> However,`our experiments only prove that large amount of data is necessary to ensure the ability of video LLMs`. We've tried to remove some of the "low-quality" datasets from the 783k training data, for example the short-caption dataset and classification dataset. However, this didn't improve the model's performance. A more systematic analysis of data quality is necessary and should be pursued in future endeavors.

---

> > ### Comment · Reviewer_tQ9n · 2024-11-27
> >
> > Thanks for responding to the comments.

---

### Meta-Review · Area_Chair_rPG6 · 2024-12-22

**Metareview:**

The manuscript received ratings of 5, 5, 5, and 6. Reviewers raiased several issues in the manuscript, including presentation issues (e.g., concepts without explanations, confounding conclusions), limited novelty (spatial pooling instead of temporal pooling and the application of LoRA weight aggregation), complexity of post-training optimization, and concerns regarding the generalizability and effectiveness of the proposed approach compared to using higher-quality SFT data or including video data during the pre-training stage. Authors provided a rebuttal to address the concerns of the reviewers, including an analysis on the difficulties of transferring from image MLLMs to video MLLMs. Post-rebuttal, 3 reviewers still remained negative mentioning the issues of limited novelty and clarity, number of heuristics in the approach, and concerns regarding the generalization and effectiveness of the proposed approach.  Given the reviewers comments, rebuttal and discussions, the recommendation is reject. Authors are encouraged to take into consideration reviewers feedback to improve the manuscript.

**Additional Comments On Reviewer Discussion:**

Reviewers raiased several issues in the manuscript, including presentation issues (e.g., concepts without explanations, confounding conclusions), limited novelty (spatial pooling instead of temporal pooling and the application of LoRA weight aggregation), complexity of post-training optimization, and concerns regarding the generalizability and effectiveness of the proposed approach compared to using higher-quality SFT data or including video data during the pre-training stage. Authors provided a rebuttal to address the concerns of the reviewers, including an analysis on the difficulties of transferring from image MLLMs to video MLLMs. Post-rebuttal, 3 reviewers still remained negative mentioning the issues of limited novelty and clarity, number of heuristics in the approach, and concerns regarding the generalization and effectiveness of the proposed approach.

---

### Decision · Program_Chairs · 2025-01-22

Reject